# Diet, Supplementation and Nutritional Habits of Climbers in High Mountain Conditions

**DOI:** 10.3390/nu15194219

**Published:** 2023-09-29

**Authors:** Ewa Karpęcka-Gałka, Paulina Mazur-Kurach, Zbigniew Szyguła, Barbara Frączek

**Affiliations:** 1Doctoral School of Physical Culture Sciences, University School of Physical Education in Krakow, Jana Pawla II 78, 31-571 Krakow, Poland; 2Department of Sports Medicine and Human Nutrition, Institute of Biomedical Sciences, University School of Physical Education in Krakow, Jana Pawla II 78, 31-571 Krakow, Poland; paulina.mazur@awf.krakow.pl (P.M.-K.); zbigniew.szygula@awf.krakow.pl (Z.S.); barbara.fraczek@awf.krakow.pl (B.F.)

**Keywords:** mountaineering, nutrition, diet, mountains, himalaism

## Abstract

Appropriate nutritional preparation for a high-mountain expedition can contribute to the prevention of nutritional deficiencies affecting the deterioration of health and performance. The aim of the study was to analyze the dietary habits, supplementation and nutritional value of diets of high mountain climbers. The study group consisted of 28 men (average age 33.12 ± 5.96 years), taking part in summer mountaineering expeditions at an altitude above 3000 m above sea level, lasting at least 3 weeks. Food groups consumed with low frequency during the expedition include vegetables, fruits, eggs, milk and milk products, butter and cream, fish and meat. The energy demand of the study participants was 4559.5 ± 425 kcal, and the energy supply was 2776.8 ± 878 kcal. The participants provided 79.6 ± 18.5 g of protein (1.1 ± 0.3 g protein/kg bw), 374.0 ± 164.5 g of carbohydrates (5.3 ± 2.5 g/kg bw) and 110.7 ± 31.7 g of fat (1.6 ± 0.5 g/kg bw) in the diet. The climbers’ diet was low in calories, the protein supply was too low, and the fat supply was too high. There is a need to develop nutritional and supplementation recommendations that would serve as guidelines for climbers, improving their well-being and exercise capacity in severe high-mountain conditions, which would take their individual taste preferences into account.

## 1. Introduction

There is a dynamic development and interest in mountain sports, including mountaineering and Himalayan mountaineering [1]. Due to the negative effects of weather and environmental conditions, mountain climbers are exposed to the risk of health deterioration and even loss of life [2]. Both short- and long-term exposure to hypoxia in high mountains causes extensive physiological changes [3]. Due to hypoxia, the partial pressure of oxygen in the blood and tissues decreases. The human body, in response to unfavorable conditions in terms of oxygen availability, responds with hyperventilation and activation of the sympathetic nervous system and causes an increase in cardiac output, which is associated with an increase in metabolic demand. Due to the conditions prevailing in the high mountain environment—i.e., reduced atmospheric pressure, leading to the development of hypobaric hypoxia, increased UV radiation, lack of access to fresh food, and thus low supply of antioxidant ingredients with food, heavy physical effort and mental stress—an increased production of reactive oxygen species (ROS) is observed, which is referred to as oxidative stress [4]. The effect of ROS in climbers returning from high-altitude expeditions is the development of many diseases, including neurodegenerative processes [5] and damage to the intestinal barrier, leading to consequences such as bacterial translocation and local/systemic inflammatory reactions [6].

At high altitude, both appetite and sense of taste are impaired. The feeling of satiety occurs after smaller meals in terms of volume, therefore smaller amounts of food are eaten. This phenomenon, called “mountain anorexia”, makes it difficult for some people to maintain body weight at an altitude of 3600 m and extremely difficult for the majority of people over 5000 m above sea level [7]. In addition, weight loss is caused by reduced energy intake due to gastrointestinal disorders [8], increased basal metabolism, altered concentrations of hormones affecting satiety, including leptin, and water loss [9,10,11,12,13,14,15]. Fluid requirements in high altitude conditions are higher than at sea level due to hyperventilation, low humidity, sweat losses [16,17] and increased diuresis as a result of downregulation of the renin-angiotensin-aldosterone system, especially during the first days of stay [18]. 

Appropriate preparation for the expedition, in particular attention to the factors that climbers have an influence on, e.g., nutrition and hydration, can contribute to the prevention of nutritional deficiencies and dehydration, affecting the deterioration of health and fitness of the body. Inadequate supply of carbohydrates during endurance activities may result in hypoglycemia, lower glycogen content and muscle fatigue [19]. On the other hand, inadequate supply of energy and protein, which is the cause of the loss of lean body mass, has a negative effect on aerobic capacity, muscle strength and immunity [20,21,22]. High-altitude hypoxia increases fluid loss and contributes to dehydration, which reduces aerobic capacity in high-altitude conditions [23]. Optimal hydration can be limited by environmental conditions, but also by the time and fuel resources needed to prepare drinking water, sourced from the glacier [24]. Reduced thirst at high altitudes is another factor contributing to dehydration [25].

Many studies have been conducted on changes in body composition caused by staying at high altitude [26,27], or approaches to feeding teams of mountaineers in the high mountains [28,29,30]. There is still a lack of precise guidelines on nutrition in high-mountain conditions, focused on the real needs and referring to the preferences of climbers. The following research hypothesis was put forward: climbers’ diets are characterized by discipline-specific inaccuracies resulting from being in high-altitude conditions. If the hypothesis of a deficient diet in high altitude conditions among Polish climbers is confirmed, this study will provide a starting point for creating dietary recommendations. This article aims to indicate what, from the point of view of Polish climbers, is the most difficult to achieve in high mountains in terms of nutrition and to provide data from the analysis of food diaries kept during the expedition.

## 2. Materials and Methods

### 2.1. Study Participants

The study group consisted of 28 men from Poland aged 23 to 42 years, taking part in mountain expeditions at least once a year during the summer period, during which they stayed at altitudes above 3000 m above sea level for at least 3 weeks. The persons qualified for the survey and supplementing the surveys were members of alpine clubs associated with the Polish Mountaineering Association. The anthropometric data and BMI of the group are presented in Table 1.

Among the subjects, 42.9% of them said they had been mountain climbing for more than 10 years, 10.7% of the subjects 9–10 years, 21.4% of the subjects 7–8 years, 17.9% 5–6 years, and 7.1% 1–4 years. The subjects had extensive climbing experience, both in rock climbing (sport climbing levels between 7b and 8c), bouldering, drytooling, and mountaineering on their own belay. Among the respondents, 82.1% train sport climbing 2 or more times a week. In addition to climbing, those surveyed (92.86%) introduce complementary training: running, biking, swimming, weightlifting, skydiving or skiing, among others.

Within the group of 28 climbers, 15 kept food diaries from three days of expedition while climbing in Peru (Cordillera Blanca) and Pakistan (Shimshal Valley). Subjects who participated in mountain expeditions, where the time spent at an altitude above 3000 m above sea level was not less than 3 weeks, were enrolled in the study. The macronutrient content of the diet was assessed in mountaineers belonging to the Polish Mountaineering Association between the ages of 22 and 40. The anthropometric data and BMI of the group are presented in Table 1. At the time when the food diaries were kept, the climbers were at an altitude of 4000–6000 m. The goal of the climbing group in Peru was to ascend, among others, the 800 m Cruz del Sur route on the rock monolith La Esfinge (5325 m). A new route was established on Ocschapalca (5888 m) and Nevado Churup (5495 m), and ascents were made on Artesonraju (6025 m) and Alpamayo (5947 m). Climbers targeting peaks in the Shuijerab Mountain Group in Pakistan carved out a route on the west face of the pristine Trident Peak (6150 m). 

Before participating in the project, the climbers had a consultation with a sports medicine doctor. On the basis of previously performed examinations (ECG, blood and urine tests), the ability to practice mountaineering, as well as to participate in the project, was assessed. The presence of chronic diseases and age over 45 were disqualifying factors from participating in the project.

The Bioethics Committee of the Regional Medical Chamber of Krakow granted approval for the research project (68/KBL/OIL/2022), which was carried out in accordance with the Declaration of Helsinki. After understanding the risks and benefits of participating in the study, all climbers gave written informed consent to participate in this study.

### 2.2. Study Design

#### 2.2.1. Nutrition Survey

The analysis of dietary habits and attitudes was based on the author’s survey. Responses to 54 questions (8 open, 46 closed) were analyzed. The questionnaire was divided into 3 parts concerning the following: anthropometric data and mountain experience and climbing experience, as well as health status and emerging ailments in the high mountains and after returning from the mountains; preparation for the expedition related to health analysis; eating behavior during the mountain action (including preferred foods, supplements, liquids, frequency of consumption of food groups in the mountains, appetite). The questionnaire was prepared on the basis of literature related to nutrition in high mountain conditions and a questionnaire to study dietary views and habits [31]. Responses to the questions on the sheet, which were analyzed, are included in the Appendix A.

#### 2.2.2. Nutritional Analysis of the Diet

The supply of selected macronutrients during the mountaineering expedition was determined by analyzing the whole-day rations obtained using the 3-day food diary method. The expedition participants were asked to note down all the foods, meals and supplements eaten, and fluids drank during the 3 active days of the expedition. These were days of acclimatization and alpine climbing on the way to the summit, and the activities noted during this time were mainly hiking with climbing elements. Climbers were given detailed instructions on how to keep food diaries before leaving for the expedition in order to minimize recording errors. The diaries contained commercially available products, so the necessary data for analysis was read from labels and nutritional tables. In the case of food products such as bars, energy gels and freeze-dried products, climbers were asked to provide the product name, manufacturer and product weight.

The data from the diaries were meticulously entered into the program Aliant Dietetic Calculator (Anmarsoft, Gdańsk, Poland; version: 85; database: 6.2) for quantitative analysis of the climbers’ diets during the active days of the expedition. The program used the “Tables of food composition and nutritional value” [32] as a database, taking into account food recipes and losses due to product processing. The average intake of macronutrients (protein, fat, total carbohydrates), dietary fiber, simple carbohydrates, saturated fatty acids, and sodium from 3 days was referred to recommendations for athletes [33,34,35,36,37].

#### 2.2.3. Anthropometric Measurements

The subjects’ body heights were measured before the expedition using a Seca 217 anthropometer (Seca GmbH & Co. KG, Hamburg, Germany) with an accuracy of 1 mm. The athletes’ body weight was determined using InBody 120 body composition analyzer (Inbody Bldg., Seoul, Republic of Korea) in the morning after a standardized meal between 7:45 and 8:30 a.m.

#### 2.2.4. Measurement of Energy Expenditure

Measurements of energy expenditure during the expedition were made by the researchers using a monitor recording heart rate Polar M430 (Polar-Electro, Kempele, Finland), which was recorded 24/7 for 3 active days. Pulsometers consisted of watches (receiver) and straps with transmitters Polar H10 sensor (Polar-Electro, Kempele, Finland), worn on the chest at the level of the gladius process in the anterior midline. The watches were placed on the hand so that they were adjacent to the body just below the wrist bone. Before the measurement began, the required data were entered into the device for each athlete: gender, age, body weight and height. Before the start of physical activity, the subjects were instructed to put on the transmitter strap, start the physical activity recording option on the device (climbing outdoors), and when the physical activity was over, to turn off the recording by pressing the corresponding button on the watch. In the case of recording activity during acclimatization or when reaching the climbing start point, the training option was not enabled. All climbers were advised not to remove the devices from their wrists for the duration of the energy expenditure recording being carried out (sleep, daytime activity, physical activity). Energy expenditure was recorded in the Polar Flow app. For those whose measurements were not taken using heart rate monitoring, an averaged physical activity index (PAL) value was calculated based on the measurements of the other climbers, which was 2.6, and then the total metabolism value was calculated by multiplying the basal metabolism value, calculated using the Harris Benedict formula, by the PAL value.

#### 2.2.5. Statistical Analysis

Analyses were carried out using Statistica 13.3 statistical package (StatSoft Inc., Tulsa, OK, USA) by determining the parameters of descriptive statistics, i.e., mean value of macronutrient intake, standard deviation (SD) and significance level (*p*). Differences between energy supply and demand were analyzed using Student’s *t*-test for related variables. A test probability of *p* < 0.05 was considered significant and the statistical significance of *p* < 0.01 was considered highly significant.

## 3. Results 

### 3.1. Survey Analysis

The results of the survey indicate that 85.7% of climbers do not perform blood tests before the expedition, in contrast to 14.3% of the surveyed group who control their health. The vast majority of respondents also do not control the level of iron (92.9%) or ferritin (96.5%). Only 7.1% of people consult a dietician before departure. Before leaving for an expedition, 75% of the respondents follow a normal diet, while 25% report using a diet other than the usual one, e.g., gluten-free, semi-vegetarian, vegan and vegetarian.

Of the climbers, 85.7% declare that their diet significantly differs from the usual diet in the mountains, while 14.3% of climbers do not report any changes in their diet in the mountains. Of the respondents, 67.9% assess their appetite in the mountains 3000 m above sea level as good, 14.3% as average, 14.3% as very good, while 3.6% of people assess their appetite as low.

The most common choices in terms of products consumed in the mountains among the respondents include chocolate bars, freeze-dried dishes, chocolate, energy gels, gummies, salty snacks, dried meat, fruit mousses, canned meat and fish, kabanos sausages, sandwiches, halva, dried fruit, nuts and instant food (jellies, puddings, soups) (Figure 1).

Cereal bars, date bars, high-protein bars and chocolate coated bars are the most popular options among climbers. Freeze-dried meals are the most frequently used nutritional solution after the end of a mountain action with bivouac (82.1%). A large percentage of climbers (75%) declare an overabundance of sweet taste, caused by the high frequency and quantity of carbohydrate snacks consumed, yet 78.6% of respondents prefer sweet snacks in the mountains as their first choice. Salty or acidic products, including meat, cheese, sausages, sandwiches and nuts are preferred by 21.4% of respondents. 

Food groups consumed during the expedition with low frequency include vegetables, fruit, eggs, milk and milk products, butter and cream, fish and meat (Figure 2).

In turn, the following products are most often consumed by climbers: sugar and sweets (85.7%), nuts and seeds (67.9%), wholegrain cereal products (64.3%) and refined cereal products (57.1%). Among the climbers surveyed, 7.1% of them realized the correct dietary recommendations regarding the frequency of consumption of vegetable oils (daily intake), 17.9% took 1–2 servings of fruit daily, 17.9% took cereal products daily, and 25% of people took fish 1–2× a week.

The freeze-dried products most often consumed by climbers during the expedition include lunch dishes, soups, porridges, oatmeal, desserts and fruit. Freeze-dried food is willingly eaten by 57.1% of the respondents, and as many as 50% of climbers declare the occurrence of digestive discomfort (flatulence, abdominal pain, diarrhea) after eating freeze-dried food. The vast majority of climbers (92.9%) declare that the amount of food affects their well-being in the mountains, and 96.4% of people emphasize that the quality of food is also very important. Despite this, as many as 57.1% of climbers do not take the right amount of food with them to the mountains. This is due to logistical errors or an attempt to minimize the weight of the backpack. Regeneration after the end of the mountain action lasts, depending on the difficulties the climbers encountered, from 1 to 7 days.

Climbers most often take isotonic drinks (78.6%), tea (60.7%), water (53.6%), coffee (14.3%) or juices (10.7%) with them in the mountains, as well as beer (3.6%), electrolyte solution (3.6%) and also kissel (3.6%). The amount of fluids consumed by climbers in the mountains is shown in Figure 3.

The respondents (57.1%) believe that the amount of fluids they drink in the mountains is not sufficient. Among the group of respondents, 85.7% of people use electrolytes as an addition to water. Only 14.3% of respondents use water purification tablets when drinking water from glaciers. 20% of the respondents experienced acute mountain sickness. As many as 94.4% of climbers claim that one of the causes of their high-altitude sickness could be inadequate hydration of the body.

Dietary supplements used by climbers include protein and protein-carbohydrate supplements, caffeine, probiotics, omega-3 fatty acids, as well as some vitamins (multivitamins, vitamin B, vitamin D, vitamin C, vitamin K2) and minerals (iron, magnesium) (Figure 4).

After returning from the expedition, 73.3% of the respondents observed a decrease in body weight. 

### 3.2. Analysis of Nutritional Value of Diet

The energy demand of the study participants was 4559.5 ± 425 kcal, and the energy supply was 2776.8 ± 878 kcal. Climbers were supplied with supplementation an average of 588 kcal, which accounted for 21.2% of their energy supply. Generally, only 2 people, i.e., 13.33% of the respondents, covered their needs in accordance with the standard (Table 2).

A highly significant (*p* < 0.01) difference was found between energy supply and demand. Consumption is much lower than the theoretical requirement.

The average energy and nutritional value of the diet obtained from the analysis of the food diaries is shown in Table 3. Most climbers (93.3%) met the carbohydrate requirement, and 46.7% met the protein requirement. All climbers met the requirements for fats, and 33.3% of them exceeded the normative values. 

## 4. Discussion

Covering energy requirements while climbing high mountains is very challenging for participants in high altitude expeditions. Our study revealed that most climbers did not meet their energy needs with diet (Table 2). This result is consistent with other studies. Kasprzak et al. [30] analyzed the diet of nine men participating in an expedition to the Alps and staying at an altitude of 3200 to 3616 m above sea level. Climbers were supplied with a diet of 1919.3 ± 771.88 kcal/day, which significantly differed from dietary recommendations for this group of physically active persons. In turn, Mariscal-Arcas et al. [29] conducted a quantitative analysis of the diet of seven climbers in an expedition base camp at an altitude of 4500 m above sea level. The average energy supply with the diet was 2833 kcal/day, which provided an insufficient supply of energy relative to demand. The results of another study showed that a group of Italians and Nepalese during an expedition to the Himalayas supplied, respectively, 2793 kcal/day and 2775 kcal/day, which was not enough in relation to the demand [28]. The dietary energy supply in our study was similar to the results presented above (Table 2). According to the recommendations and the level of physical activity, the energy demand of athletes may range from 40 to 70 kcal/kg bw/day (2000–7000 kcal/day for a 50–100 kg athlete) [37]. It has also been shown that estimated energy expenditure increases with increasing climb angle by 1.5–2.0 additional kcal/min for climbs of 80–90° and 5 additional kcal/min for climbs of 102° [38]. Mountain experience can affect energy expenditure. Energy expenditure of experienced climbers during mountain climbing was found to be lower [39].

Increased energy expenditure during long high-altitude hikes may be responsible for a negative energy balance leading to changes in body composition, namely loss of fat mass and lean body mass [8,11,40]. Analysis of energy expenditure conducted using the double labeled water method with Mount Everest climbers indicate that energy expenditure during physical activity is 1.85–3.0 times higher than resting energy expenditure at sea level [11]. Negative energy balance is the main cause of muscle protein catabolism. After depletion of glycogen and fat stores, protein stores are catabolized to cover energy expenditure [41]. Climbers lose at least 3% of their body weight after eight days at 4300 m and 15% after three months at 5300–8000 m [42]. The results of a survey we conducted also showed weight loss in 73.3% of climbers returning from an expedition. Loss of lean body mass negatively affects aerobic capacity [20], muscle strength [21], immune functions of the body [22] at high altitudes, which may increase the risk of disease and injury in these extreme conditions. However, muscle breakdown can be beneficial as an adaptation in high altitude conditions because it increases the density of capillaries relative to the muscle cell [43]. The density of capillaries within a given area of muscle tissue will greatly contribute to matching the delivery of oxygen and nutrients with the myofibers’ metabolic needs, particularly during contractile activity [44]. 

Mountaineers need to provide carbohydrate sources with their diet during high-altitude physical activity to maintain body weight, recover adequately and replenish glycogen stores [45,46]. At sea level, carbohydrate intake before and during prolonged moderate-to-high-intensity physical activity conserves endogenous carbohydrate stores and delays the onset of fatigue [47,48]. Nutritional recommendations for people staying at high altitudes say that carbohydrate intake should account for at least 60% (6–8 g/kg bw/day) of total energy intake [49,50]. These guidelines are based on research showing that total carbohydrate oxidation during high-altitude exercise is greater than exercise of the same intensity at sea level [51], which leads to a faster use of carbohydrate stores in the body (i.e., muscle and liver glycogen and blood glucose). In our study most climbers met the carbohydrate requirement (Table 3). The carbohydrate supply was similar to the results reported by Bondi et al. [28], describing the nutrition of Italian trekkers during an expedition to the Himalayas. The Nepalese porters consumed larger amounts of carbohydrates (7 ± 2 g carbohydrates/kg bw/day) [28]. In a study by Mariscal-Arcas et al. [29] the level of carbohydrates in the diet of climbers was too low (39.5%). The climbers in our study provided more simple sugars with their diet (Table 3) than the group of Italians and Nepalese in the study by Bondi et al. [28] (45 and 90 g/day, respectively). While total daily carbohydrate requirements may be elevated when staying at high altitudes, it is unclear whether additional carbohydrate intake provides an ergogenic performance benefit. At high altitude, the potential ergogenic effect of carbohydrate supplementation on exercise performance may be partially modulated by the acclimatization state [52,53,54]. Bradbury et al. [55] showed that carbohydrate supplementation during steady-state exercise does not improve exercise performance in lowland residents acutely exposed to hypoxia or residing at high altitude (4300 m) for 22 days.

An adequate supply of fiber with the diet is a major challenge for participants in high altitude expeditions due to limited access outside base camp. The results of the Bondi et al. [28] study showed that the Nepalese porter group provided 23 g/day of fiber, while the Italian trekker group provided 43 g/day. The results of our study indicate similar fiber supply values to the Nepalese group (Table 3). The fiber supply provided with the diet of the mountaineers in the Kasprzak et al. [30] study was even lower (12.2 g/day). Low intake of dietary fiber and resistant starch may lead to decreased bowel movements resulting in decreased bowel function, and also decreased the diversity of gut microbiota [56].

A high-carbohydrate, low-fat diet at altitude increases the respiratory quotient (RQ). If only fats are used for energy production, this coefficient is 0.7, while using carbohydrates (or proteins) increases to a value close to 1. The effect of this change in RQ is to increase the partial pressure of oxygen, resulting in an increase in arterial blood oxygen saturation [43,57]. Carbohydrate utilization is also a more efficient process in terms of energy yield per unit of O_2_ consumed [58]. In the high mountains, a high-fat diet can lead to chronic muscle fatigue due to insufficient amounts of readily available carbohydrates. Diets rich in fatty acids require more oxygen during metabolism, slowing down acclimatization [59]. In our study all climbers met the requirements for fats and part of them exceeded the normative values (Table 3). The dietary fat supply of the climbers from our study is similar to the dietary fat supply (38%) of the Nepalese porters studied by Bondi et al. [28]. The percentage of fats in the diet during the expedition was lower among the Italians trekkers (24%) [28]. In a study by Mariscal-Arcas et al. [29] the level of fat in the diet of climbers was too high (45.5%). The diet prepared by the Sherpas at the base during the expedition was characterized by a high content of saturated fatty acids (derived mainly from meat, butter and cheese) [29]. The supply of saturated fatty acids provided with the diet of the mountaineers in our research was also high (Table 3). Specialists from the Academy of Nutrition and Dietetics, Dietitians of Canada, American College of Sports Medicine (ACSM) and the International Society of Sports Nutrition (ISSN) recommend that fat intake by athletes should be in line with public health guidelines [36,37]. According to the recommendations of the European Food Safety Authority (EFSA), saturated fatty acids intake should be as low as is possible within the context of a nutritionally adequate diet [35].

An adequate supply of protein should be provided with the diet during physical activity at high altitudes to prevent weight loss and to ensure an adequate amount of this nutrient for tissue construction and repair [45]. The dietary protein intake necessary to support metabolic adaptation, repair, remodeling and protein turnover typically ranges from ISSN 1.2 to 2.0 g/kg bw/day to prevent loss of lean mass (FFM) [36], which is confirmed by the guidelines, which give a range of 1.7–2.2 g/kg bw/day [37]. Protein intake, especially branched-chain amino acids, is crucial for regulating muscle protein synthesis [60]. In the study by Bondi et al. [28], a group of Italians and Nepalese during an expedition to the Himalayas provided 1.2 ± 0.4 g of protein/kg of body weight, respectively, and 1.3 ± 0.3 g protein/kg bw/day. These results are similar to those we presented (Table 3). In contrast in a study by Mariscal-Arcas et al. [29] climbers were taking in 1.5–2.5 g protein/kg body weight, which is in line with the recommendations for athletes.

Increasing the protein content in the diet during caloric restriction to 1.6 g/kg bw/day improves nitrogen balance and allows for maintaining lean body mass [61]. These assumptions are particularly important for athletes who often need to prevent the loss of lean body mass. This approach was tested during a stay at high altitude [62]. However, protein increases the feeling of satiety [63], which is not a beneficial effect at high altitude, where climbers face high-altitude anorexia [64]. A diet richer in protein may also not align with the dietary preferences of high-altitude climbers. There are no studies that would evaluate the effect of protein supplementation on the feeling of satiety at high altitude. The most practical strategy to protect the body from lowering lean body mass at altitude seems to be to stimulate MPS and/or reduce proteolysis with a low-volume protein supplement rich in branched-chain amino acids that promote MPS, especially leucine [27]. A high-protein diet (2 g protein/kg bw/day) did not spare lean body mass in 4300-metre-altitude military personnel because doubling protein intake (compared to the standard diet group) resulted in a parallel increase in whole-body protein oxidation, suggesting that the additional protein consumed in the high-protein group was mainly used as an energy source during the 21 days of energy deficit at high altitude [65]. Proper nutrition can offset the loss of muscle mass, but it is not possible to completely prevent muscle atrophy [27]. 

The increased loss of breathing water and diuresis often seen in the early response to altitude exposure can result in a significant increase in water requirements, while reduced thirst and changes in fluid availability in a new environment can alter habitual drinking practices [45,66,67]. According to ACSM experts, the daily requirement for fluids is 4 to 5 L during altitude training and competition [36], while the ISSN encourages individual monitoring of hydration status to determine athletes’ fluid needs [37]. At moderate altitudes up to 4000 m above sea level, water loss from the respiratory tract can be increased to 1900 mL/day in men [45] and in addition, urinary excretion can be increased by 500 mL/day [59]. Our results show that nearly half of the respondents drink 2–3 L of fluids per day in the mountains, while the rest of the respondents take in even smaller amounts of fluids per day (Figure 3), which can lead to dehydration and have a negative impact on acclimatization. A study by Ladd et al. [24] based on two methods of measuring hydration status (urine specific gravity and ultrasound measurements of the inferior vena cava size and collapsibility index) found that about half of the surveyed climbers who had climbed Denali in Alaska were dehydrated. Water supply was monitored in a group of Italians and Nepalese during an expedition to the Himalayas, reaching 3099 ± 462 g/day and 3240 ± 310 g/day, respectively [28]. A common custom of the group of Italians and the group of Nepalese porters participating in the Himalayan expedition was to consume tea frequently throughout the trek, more than once a day [28]. The study conducted by Scott et al. [68] shows that even when drunk at high altitude where fluid balance is stressed, there is no evidence that tea acts as a diuretic when consumed through natural routes of ingestion by regular tea drinkers, but that it does have a positive effect on mood. Only 14.3% of respondents in our study use water purification tablets when drinking water from glaciers. Available ice and snow may be contaminated and unfit for consumption. Giardia lamblia, an intestinal parasite that causes diarrhea, is found in high-altitude regions [69]. In the absence of water treatment tablets, there is a high risk of parasitic infection. 

The use of supplementation by climbers in high mountains can improve exercise capacity and recovery, concentration, provide deficient dietary components and also have a positive effect on immunity. The most commonly used supplement by climbers in our survey is caffeine (Figure 1). The ergogenic effect of caffeine in sports has been demonstrated in many scientific studies. Caffeine influences cognitive and physical functions by blocking adenosine A1 and A2a receptors in the central nervous system and peripheral tissues. Doses of 1–4 mg/kg body weight improve alertness, concentration, and reaction time [70], while doses of 3–6 mg/kg bw caffeine can enhance cognitive performance, motor skills, and physical performance in many types of sports [71,72]. These features are especially important in technically demanding places when climbing and when descending mountains or rappelling. Another common dietary supplement chosen by climbers in our survey is protein supplement (Figure 1). The use of a protein supplement rich in leucine seems to be a reasonable solution to protect the body against the reduction of lean body mass at high altitude [27]. Other dietary supplements used by climbers in our study include protein-carbohydrate supplements, probiotics, omega-3 fatty acids, as well as some vitamins (multivitamins, vitamin B, vitamin D, vitamin C, vitamin K2) and minerals (iron, magnesium). Overall, a positive association between omega-3 fatty acids supplementation and reaction time, skeletal muscle recovery, inflammatory markers and cardiovascular dynamics was reported [73]. Supplementation has also been shown to influence muscle protein synthesis, especially in conditions such as immobilization and energy restriction, or when consumed with other nutrients [74,75]. Substances with antioxidant activity, e.g., vitamin C, α-tocopherol and α-lipoic acid have a modulating effect on oxidative stress and AMS symptoms at high altitudes [76]. In order to reduce the frequency of upper respiratory tract infections, improve intestinal barrier function and reduce inflammation, probiotic supplementation is recommended [37]. Current evidence points to a potential improvement in endothelial function at high altitude after NO_3_ supplementation [77]. Beetroot juice (and more precisely the nitrates it contains) is not a good prevention of AMS. Nitrates are not recommended as a prophylactic or ergogenic agent for already existing AMS [78].

A decrease in appetite is most often observed in the first few days after arriving at altitude. Staying at an altitude below 5000 m above sea level after a few days of acclimatization, the appetite increases again, while at extreme altitudes the phenomenon of high-altitude anorexia is particularly acute [43,79]. Our results show that the majority of respondents assess their appetite in the mountains 3000 m above sea level as good. It is worthwhile for climbers to also manage their appetite and select snacks in the mountains according to their own preferences and gastrointestinal tolerance. Karl et al. [80] in their study proved that exposure to high altitude was closely related to the preference for sweet foods and the preference for low-protein foods over higher-protein foods. These results suggest that sweet foods low in protein may be particularly appealing during the initial period at high altitude. After weight loss and acclimatization, study participants preferred high-fat foods [80]. In our research, a large percentage of climbers report an excess of sweet taste due to the high frequency and quantity of carbohydrate snacks consumed. However, there is still a majority of respondents (78.6%) that mostly prefer sweet snacks in the mountains. The Aeberli et al. [81] study examined changes in appetite and food preference after a rapid ascent to 4559 m with a starting altitude of 446 m. Liking of all food items except for sweet food was reduced after a meal at baseline and at high altitude. There was a significant increase in palatability of spicy foods at high altitude and an interaction between food preference and susceptibility to acute mountain sickness (AMS). Subjects that develop AMS seem to prefer sweet foods over savory foods [81]. In our study, salty or acidic products, including meat, cheese, sausages, sandwiches, and nuts are preferred by 21.4% of respondents. Reducing the available amount of sodium in humans leads to an increase in salt palatability [82], and high-altitude climbing has been shown to lead to sodium diuresis associated with suppression of the renin-angiotensin-aldosterone system [6]. Our results show that climbers are most likely to consume both sweet and salty snacks and dried or freeze-dried products in the mountains (Figure 1). The freeze-dried products most often consumed by climbers during the expedition include lunch dishes, soups, porridges, oatmeal, desserts and fruit. As many as half of climbers report experiencing digestive discomfort (bloating, abdominal pain, diarrhea) after eating freeze-dried food. Gastrointestinal complaints are frequently reported during high altitude climbing (> 2500 m above sea level), although their etiology is unknown. Initial data also suggest that prolonged hypoxic exposures can compromise the intestinal barrier through alterations in immunological function, microbiota, or mucosal layers [83].

Among the climbers surveyed in our study, 7.14% followed the correct dietary recommendations regarding the frequency of consumption of vegetable oils (daily intake), 17.85% took in 1–2 servings of fruit per day, 17.85% took in cereal products several times a day, and 25% took in fish 1–2× per week with reference to the Swiss food pyramid for athletes [84]. None of the climbers followed the recommendations for adequate frequency of consumption of vegetables and dairy products. Vegetables and fruits are the main and very important sources of vitamins and minerals in a diet of athletes. However, consuming these products outside of base camp is difficult to accomplish. Taken by mountaineers on the summit attack or used during climbing, it should be as light as possible and easy to prepare, and additionally it should not freeze. It is worth considering the use of vegetables and fruits in the form of lyophilisates. Especially in the base camp, where the availability of products may be greater, it is worth taking care of the supply of these products.

Our research shows that the vast majority of climbers do not perform blood tests before an expedition, and do not monitor their iron or ferritin levels. At high altitude, the process of erythropoiesis is intensified, which increases the need for iron [85]. Therefore, even before the expedition, it is worth taking blood tests and consulting the result with a sports medicine doctor who would help choose the right dose of the supplement [86]. Current recommendations suggest assessing the iron level 8–10 weeks before the start of activity in high mountains [87] and starting appropriate oral supplementation 2–3 weeks before exposure to altitude and continuing this supplementation [88,89,90]. Of the climbers surveyed, only 7.1% of them work with a nutritionist before an expedition. Evidence-based nutritional strategies and a nutrition plan provided by a dietitian can lead to improved performance and recovery [91]. 

The results of the present study may have significance in the cooperation of sports nutritionists and trainers with climbers, allowing them to prepare more effectively for high altitude expeditions. The results obtained may also serve as practical guidance for high altitude climbers themselves.

### Limitations and Directions for Future Research

We acknowledge that this study has several potential limitations. In the next such study, it would be worthwhile to expand the analysis of the energy value of the diet to include days at base camp (rest days) and to increase the group of subjects. However, it should be noted that the study group was homogeneous in terms of mountain goals and level of climbing, so finding more subjects is a challenge. It would be worthwhile for a nutritionist to also participate in the expedition, so that he or she is in constant contact with the cooks responsible for preparing the meals and has control over the amount of food consumed. We also recommend that future studies analyze the micronutrient content of foods used by climbers so that the analysis is more accurate. In addition, it is worth focusing on analyzing the supplements used by climbers in the high mountains and their effects on health, mood and physical performance.

## 5. Conclusions

The climbers’ diet was low in calories in relation to the high energy requirements, the protein supply was too low, and the fat supply was too high. Mountain climbers should aim to increase the supply of carbohydrates.During climbing, climbers usually take snacks (bars, energy gels, gummies), which are a great supplement to carbohydrates. Providing adequate fiber outside base camp during a climb can be a problem, so especially at base camp, where food availability is greater, it is worthwhile to ensure a supply of foods abundant in this dietary component.The use of a protein supplement at high altitudes seems to be a practical strategy that can reduce the loss of lean body mass. There are many supplements with well-documented effects in sports (e.g., beetroot juice, caffeine, probiotics, omega-3 fatty acids, essential minerals and vitamins) that are worth considering as a supplement to the diet.There is a need to develop nutritional and supplementation recommendations that would serve as guidelines for climbers, improving their well-being and exercise capacity in severe high-mountain conditions, which would take into account their individual taste preferences.

## Figures and Tables

**Figure 1 nutrients-15-04219-f001:**
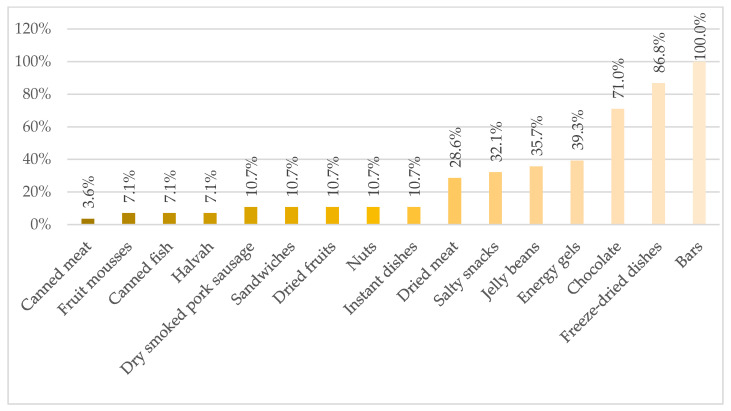
Food products most often taken on a mountain action with a bivouac above the base camp by climbers (percentage of climbers).

**Figure 2 nutrients-15-04219-f002:**
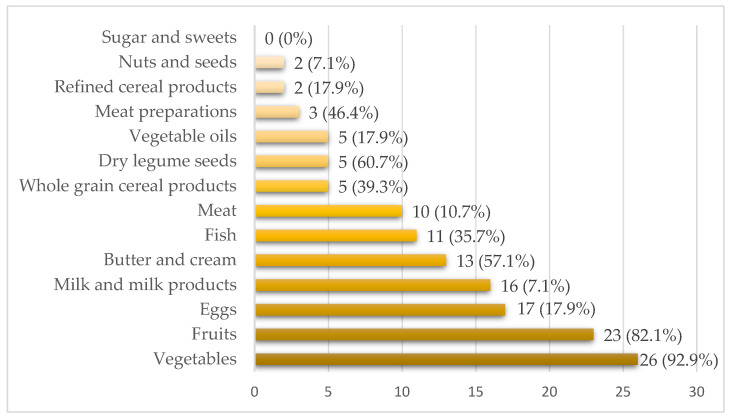
Declarations (percentage of climbers) concerning food groups consumed in high mountain conditions with low frequency.

**Figure 3 nutrients-15-04219-f003:**
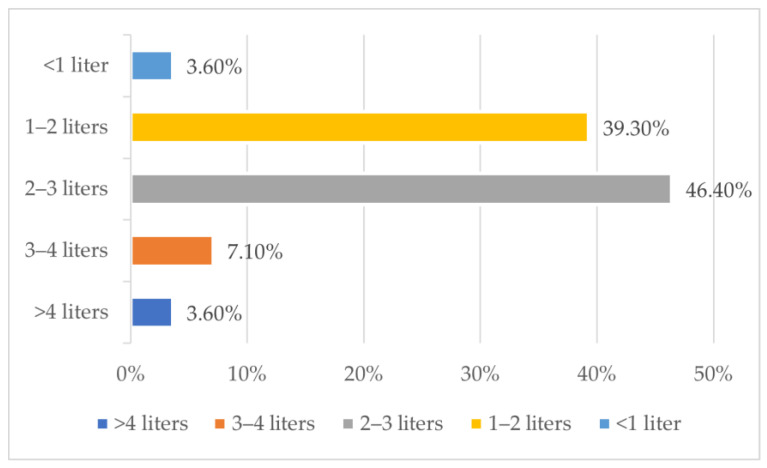
The amount of fluids consumed by climbers in the mountains (per day).

**Figure 4 nutrients-15-04219-f004:**
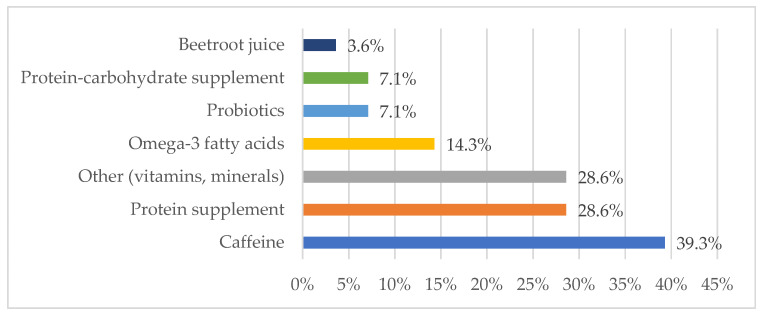
Percentage of climbers using dietary supplements during the expedition.

**Table 1 nutrients-15-04219-t001:** Anthropometric data and BMI of climbers participating in the survey (*n* = 28) and supplementing food diaries (*n* = 15).

	Men (*n* = 28)	Men (*n* = 15)
	$\bar{x}$	SD	$\bar{x}$	SD
Age [years]	33.12	5.96	29.6	5.25
Body height [cm]	179.04	6.96	179.87	7.37
Body weight [kg]	71.48	5.43	71.39	6.80
BMI [kg/m^2^]	22.31	1.37	22.06	1.61

Abbreviations: x¯—mean; SD—standard deviation; BMI—body mass index.

**Table 2 nutrients-15-04219-t002:** Differences between energy supply and demand analyzed with the Student’s *t*-test for related variables.

	Dietary Energy Supply [kcal]	Energy Requirements [kcal]
Arithmetic mean	2776.84	4559.50
Standard error	234.67	113.60
Standard deviation	908.87	439.96
−95% CI (confidence interval)	2273.53	4315.86
+95% CI	3280.15	4803.14
Mean of differences	−1782.66	
Standard error	262.4959	
Standard deviation	1016.6423	
−95% CI	−2345.6578	
+95% CI	−1219.6623	
T statistic	−6.7912	
Degrees of freedom	14	
Two-sided *p*-value	<0.0001	

**Table 3 nutrients-15-04219-t003:** Energy and nutritional value of the daily food ration of mountain climbers in relation to the recommendations for athletes [34,35,36,37].

	Nutritional Recommendations for Athletes	Diet of Climbers during the Expedition	
		$\bar{x}$	SD
Energy [kcal]	Daily energy expenditure	2776.8	878
Protein [%]	15–20 ^1^	12.1	2.8
Protein [g]	-	79.6	18.5
Protein [g/kg bw]	1.2–2.2 ^1^	1.1	0.3
Carbohydrates [%]	45–65 ^2^	52.7	8.1
Carbohydrates [g]	-	374.0	164.5
Carbohydrates [g/kg bw]	6–12 ^3^	5.3	2.5
Simple carbohydrates [g]	-	147.3	67.1
Fiber [g]	38 ^2^	24.8	8.2
Fats [%]	20–35 ^2,3,4^	36.8	7.2
Fats [g]	-	110.7	31.7
Fats [g/kg bw]	0.5–1.5 ^1^	1.6	0.5
Saturated fatty acids [%]	ALAP ^4^	13.8	5.4
Saturated fatty acids [g]	ALAP ^4^	40.5	16.5
Sodium [mg] (AI)	1500–>10,000 ^2^	3177.0	855.7

^1^—[37]; ^2^—[34]; ^3^—[36]; ^4^—[35]. Abbreviations: $\bar{x}$—mean; SD—standard deviation; bw—body weight; AI—Adequate Intake; ALAP—as low as possible.

## Data Availability

Not applicable.

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
