# Peer review of "Diet, Supplementation and Nutritional Habits of Climbers in High Mountain Conditions"

_nutrients, 2023, doi:10.3390/nu15194219_

Round 1
Reviewer 1 Report
The paper presented for review: „Diet, supplementation and nutritional habits of climbers in high mountain conditions” is a research work with a classic layout. The authors analyze dietary habits, supplementation, and nutritional value of diets of high mountain climbers.
The paper needs minor revision:
The Introduction is too long and contains much basic information about the body's physiological reaction to exposure to hypoxia. The Authors are writing about nutrition during high-altitude expeditions but only about dehydration caused by insufficient fluid intake. They should notice that higher altitudes are associated with dehydration due to increased urine output, dryer air, and more rapid breathing, resulting in a more significant loss of body fluids.
Lines 86-87: The Authors cite the article by Kasprzak et al. (33), in which study participants were climbing in the Alps. Please remove this citation or "of Himalayas."
Materials and methods
Lines 99 and 116-117: Which association's name is correct?
Line 104: Please write „mean” instead of „average”.
Lines 113 and 150: What do Authors mean by writing "three active days"? Please clarify if there were three days of just hiking or summit attack.
Lines 157-158, 167, 171-173 and 190-191: Please provide the exact details of the equipment manufacturer, including city and country.
Lines 185-188: Please provide the obtained PAL value.
Line 190: Please correct the name of the statistical package manufacturer from StatSoY to StatSoft.
Lines 192-193: Why did the Authors use the t-Student test? Did they check the normality of the data distribution?
Line 194: Please write „the statistical significance” instead of „test probability”.
Results
Figure 2: Please change commas in percentage values to dots.
Line 280: Please use „recommendations” instead of „standards”.
Line 281: There is an error caused by the citation manager.
Table 3: Please use „nutritional recommendations for athletes” instead of „Nutritional norms for athletes”. Please clarify all abbreviations used in the table.
Discussion
The discussion is very extensive, with numerous repetitions and language errors. It is not fluent to read. I recommend rewriting it. In this section, the Authors describe the results of the research. They should not show and repeat the values of the obtained results but only provide a reference to the figure or table. The conclusions are mainly a summary of the obtained results.
Lines 286, 288, 400, 411 etc.: The number of citations should be placed directly after the author's name, e.g., Mariscal-Arcas et al. [32].
Lines 313 and 316: I think that „men consume" sounds better than „men supply”.
Lines 350 and 499: There should be a subscript.
Lines 385, 407,504: Please use „our study” or „our research” instead of „our own research” or „ in own study”.
Lines 396-403: Why are the Authors writing about using diets other than usual? What can be the consequences of it?
Lines 455-456: Abbreviations appear for the first time in the text; please clarify them.
Lines 459-460: " L " should be a volume unit.
Lines 467-468: Please remove the year of the cited article.
Line 474: There is an error caused by the citation manager.
Lines 479-485: I find this information too detailed. Please remove or shorten this fragment.
Author Response
Dear Reviewer,
Thank you very much for reviewing our manuscript and all your comments.
Below are the responses to your comments.
Introduction
Thank you for your valuable comments on the introduction. We have made the following changes: the introduction has been shortened in the section on physiological adaptations of the body to hypoxia. We have added information on fluid loss and dehydration. We have added a research hypothesis at the end of the introduction.
Lines 86-87: The Authors cite the article by Kasprzak et al. (33), in which study participants were climbing in the Alps. Please remove this citation or "of Himalayas."
Thank you for this comment. The suggested change has been made, the phrase "of Himalayas" has been removed.
Materials and methods
Lines 99 and 116-117: Which association's name is correct?
We have corrected. The correct name is: Polish Mountaineering Association
Line 104: Please write „mean” instead of „average”.
We have corrected.
Lines 113 and 150: What do Authors mean by writing "three active days"? Please clarify if there were three days of just hiking or summit attack.
These were days of acclimatization and alpine climbing on the way to the summit, and the activities noted during this time were mainly hiking with climbing elements. We added this information to the manuscript.
Lines 157-158, 167, 171-173 and 190-191: Please provide the exact details of the equipment manufacturer, including city and country.
The exact details of the equipment manufacturer, including the city and country have been added.
Lines 185-188: Please provide the obtained PAL value.
The value of the physical activity factor was 2.6, the information added.
Line 190: Please correct the name of the statistical package manufacturer from StatSoY to StatSoft.
We have corrected.
Lines 192-193: Why did the Authors use the t-Student test? Did they check the normality of the data distribution?
We used Student's t-test for related variables because it is the test that have to be performed to answer the question "was the intake different from the demand?". Yes, we checked the normality of the data distribution.
Line 194: Please write „the statistical significance” instead of „test probability”.
Thank you for this comment. The suggested change has been made.
Results
Figure 2: Please change commas in percentage values to dots.
Thank you for your comment. We have changed the separators in all the charts.
Line 280: Please use „recommendations” instead of „standards”.
We have corrected.
Line 281: There is an error caused by the citation manager.
We have corrected.
Table 3: Please use „nutritional recommendations for athletes” instead of „Nutritional norms for athletes”. Please clarify all abbreviations used in the table.
We have corrected. The abbreviations used in the table are explained.
Discussion
The discussion is very extensive, with numerous repetitions and language errors. It is not fluent to read. I recommend rewriting it. In this section, the Authors describe the results of the research. They should not show and repeat the values of the obtained results but only provide a reference to the figure or table. The conclusions are mainly a summary of the obtained results.
Thank you for your comment on the discussion. The discussion has been rewritten. Repetitions have been removed and errors corrected. Appropriate references to tables and figures have been added. Information on the limitations and direction of further research has also been expanded.
Lines 286, 288, 400, 411 etc.: The number of citations should be placed directly after the author's name, e.g., Mariscal-Arcas et al. [32].
A change in citation has been made in all places in the text.
Lines 313 and 316: I think that „men consume" sounds better than „men supply”.
We have corrected.
Lines 350 and 499: There should be a subscript.
We have changed.
Lines 385, 407,504: Please use „our study” or „our research” instead of „our own research” or „ in own study”.
Thank you for this comment. We have corrected.
Lines 396-403: Why are the Authors writing about using diets other than usual? What can be the consequences of it?
Thank you for your comment. This piece of text does not constitute the essential substantive thread of the work, and there is no basis for analyzing the consequences of unconventional diets. The excerpt has been removed from the discussion.
Lines 455-456: Abbreviations appear for the first time in the text; please clarify them.
Thank you for your comment. We have removed the abbreviation and added the full name.
Lines 459-460: " L " should be a volume unit.
We have corrected.
Lines 467-468: Please remove the year of the cited article.
We have removed.
Line 474: There is an error caused by the citation manager.
We have corrected.
Lines 479-485: I find this information too detailed. Please remove or shorten this fragment.
Thank you for your comment. We have removed.
All posted changes in the text are highlighted in red.
Reviewer 2 Report
The authors propose an article analysing the dietary patterns, supplementation, and nutritional value of the diets of 18 high-altitude mountaineers. It is an intriguing study, so I will attempt to provide my perspective and make some suggestions:
- The introduction focuses primarily on the most significant references on the topic, but the hypothesis should be considered for inclusion.
- For a sharper focus, the authors should consider incorporating new practical implementations for the future, as well as separating a limitation point.
In general, it is a well-written manuscript, but it is unfortunate that the number of climbers is so small; if they can increase the previous ratings, it will be an intriguing study.
Author Response
Dear Reviewer,
Thank you very much for reviewing our manuscript and all your comments.
Below are the responses to your comments.
- The introduction focuses primarily on the most significant references on the topic, but the hypothesis should be considered for inclusion.
Thank you for your valuable comments on the introduction. We have made the following changes: the introduction has been shortened in the section on physiological adaptations of the body to hypoxia. We have added information on fluid loss and dehydration. We have added a research hypothesis at the end of the introduction.
- For a sharper focus, the authors should consider incorporating new practical implementations for the future, as well as separating a limitation point.
Thank you for the suggestion. The paragraph on limitations has been extracted and expanded. The discussion has been rewritten.
All posted changes in the text are highlighted in red.